A real-time feeding behavior monitoring system for individual yak based on facial recognition model

Yang Yuxiang
Liu Meiqi
Peng Zhaoyuan
Deng Yifan
Gu Luhui
Peng Yingqi pengyingqi@sicau.edu.cn
College of Mechanical and Electrical Engineering, Sichuan Agricultural University , Ya’an , Sichuan , China
Yang Jiachen
Electronic publication date: 2024 Oct 24
Publication date: 2024
Volume: 10
Electronic Location ID: e2427
Received 2024 May 29; Accepted 2024 Sep 26
Copyright: ©2024 Yang et al.
Copyright year: 2024
Copyright holder: Yang et al.
License: This is an open access article distributed under the terms of the Creative Commons Attribution License, which permits unrestricted use, distribution, reproduction and adaptation in any medium and for any purpose provided that it is properly attributed. For attribution, the original author(s), title, publication source (PeerJ Computer Science) and either DOI or URL of the article must be cited.
License URL: https://creativecommons.org/licenses/by/4.0/

Keywords: YOLO, StrongSORT, Individual feeding behavior, Yak, Yak face classification

Funding: National Key R&D Program of China 2021YFD1600200 program of Sichuan Agriculture University T202107 This study was supported by National Key R&D Program of China (Grant Number 2021YFD1600200). The study was also supported by subject double support program of Sichuan Agriculture University (Grant Number T202107). The funders had no role in study design, data collection and analysis, decision to publish, or preparation of the manuscript.

==============================
Feeding behavior is known to affect the welfare and fattening efficiency of yaks in feedlots. With the advancement of machine vision and sensor technologies, the monitoring of animal behavior is progressively shifting from manual observation towards automated and stress-free methodologies. In this study, a real-time detection model for individual yak feeding and picking behavior was developed using YOLO series model and StrongSORT tracking model. In this study, we used videos collected from 11 yaks raised in two pens to train the yak face classification with YOLO series models and tracked their individual behavior using the StrongSORT tracking model. The yak behavior patterns detected in trough range were defined as feeding and picking, and the overall detection performance of these two behavior patterns was described using indicators such as accuracy, precision, recall, and F1-score. The improved YOLOv8 and Strongsort model achieved the best performance, with detection accuracy, precision, recall, and F1-score of 98.76%, 98.77%, 98.68%, and 98.72%, respectively. Yaks which have similar facial features have a chance of being confused with one another. A few yaks were misidentified because their faces were obscured by another yak’s head or staff. The results showed that individual yak feeding behaviors can be accurately detected in real-time using the YOLO series and StrongSORT models, and this approach has the potential to be used for longer-term yak feeding monitoring. In the future, a dataset of yaks in various cultivate environments, group sizes, and lighting conditions will be included. Furthermore, the relationship between feeding time and yak weight gain will be investigated in order to predict livestock weight.

Introduction

The yak (Bos grunniens) is a significant species living on the Qinghai-Tibetan Plateau and in high alpine areas. Around 95% of the world’s yaks are located in China, where they are distributed across the ecological conditions of low atmospheric oxygen, a lack of grass in the winter, and year-round cold temperatures (Gao et al., 2023). In recent years, some livestock farms have bred yaks in pens by improving living conditions and feeds in an effort to increase breeding and reproduction efficiency as well as animal welfare. Feeding is a key indicator of livestock welfare, health, and productivity. Previous research has shown that continuous monitoring of cattle feeding variables can help understand the effects of diet on digestive function and performance (Wagner et al., 2020; Rial et al., 2023a; Rial et al., 2023b). Individual livestock behavior monitoring in the case of feedlot animals could inform effective feed management for these steers in the feedlot.

Traditionally, feeding behavior is visually assessed through video recordings or detectable observations but these methods are time consuming and labor intensive. To date, multi-sensors and deep learning models currently offer dependable solutions for precise livestock management. By observing changes in the behavior of the cattle, corresponding important physiological and cultivation status can be tracked. Previous studies have fitted accelerometers and inertial measurement unit (IMU) sensors to the necks, ears, noses, and legs of cattle in order to track their behavior (Peng et al., 2019; Peng et al., 2020; Liu et al., 2023; Zhang et al., 2023; Serviento et al., 2024; Weinert-Nelson et al., 2024). Moreover, machine learning techniques that combine vision have also been used more recently under conditions of continuous and real-time animal monitoring. In addition, several studies trained images of livestock including yaks using deep learning algorithms to monitor and identify behaviors such as feeding, lying, standing and walking (Sun et al., 2023) and extract body size parameters (Peng et al., 2024). Tracking objects and identifying different cow behavioral patterns are made possible by dynamic analysis. Self-protective behavior, estrus, and behavior linked to hoof sickness can all be detected using a monitoring and classification system (Zheng et al., 2023; Wang et al., 2024b; Russello et al., 2024; Li et al., 2024).

For the feeding behavior monitoring, three-axis accelerometers and/or IMU sensors have been mounted to the animal’s neck and jaws for the purpose of behavior data collecting (Norbu et al., 2021; Li & Chai, 2023; Zhang et al., 2023). Another method involved placing a microphone near the animal’s mouth to recognize the sounds of the animal feeding (Martinez-Rau et al., 2022; Peng et al., 2023; Ferrero et al., 2023) and machine learning method such as hidden Markov model-based electronic system with a microcontroller was used to run a recognition algorithm (Milone et al., 2012). These methods have been effective in recognizing feeding behaviors of livestock from other behavior patterns. However, each individual’s disposition leads them to raise in a particular feeding habit as they grow. Existing methods for tracking individual livestock feeding behavior have rarely been proposed. Therefore, these feeding behavior monitoring methods without livestock authentication function make the feeding behavior duration and distribution statistics difficult to realize in large-scale farming. Furthermore, only detecting if the cow’s head is in the feeding trough does not provide enough evidence to accurately monitor if the yaks are chewing the feed. According to our observation and previous research, behavior patterns including feeding, ruminating, competing and other observation can be detected when the cow’s head is in the trough (Reyes et al., 2024).

In this study, we develop an individual yak feeding behavior monitoring method using YOLO series and StrongSORT models. The individual feeding monitoring realized through yak’s face features and position information. For more precise explanation and detection of yaks’ behavior in trough, this study further classified general feeding behavior into feeding and picking. Finally, the distribution and duration of feeding and picking behavior were computed and shown. This work provided a monitoring model for continuously individual yak feeding behavior, which can provide useful data for precise feed management and alarms for digestive disease in particular animals. In summary, the main contributions of this paper are as follows:

•The yak behavior patterns detected in trough range were defined as feeding and picking.

•YOLO series model and StrongSORT were used to develop a real-time detection model for individual yak feeding and picking behavior based on yak face classification.

•The SEAM module was utilized in YOLO series model to enhance the yak classification performance in occlusion condition.

•The duration of feeding and picking behaviors in one day was calculated automatically based on individual behavior detection.

Materials and Methods

Location and animals

From September 12, 2022, to September 18, 2022, the experiment was conducted on a livestock farm in Sichuan, China. Eleven one-year-old yaks were raised separately in two groups, with five or six yaks housed in each group’s pen (width is 5,000 mm and length is 8500 mm), and videos were consistently collected from these yaks. In the pen’s resting area, soft sands and wood dusts have been used as bedding. The yaks were raised in concrete feed troughs with unrestricted access to water. The Chinese Ministry of Agriculture published the NYT/815/2004 dietary standard for dry matter intake, net energy, crude protein, and minerals for feeding beef cattle. The experiment was assessed by the Sichuan Agricultural University’s Animal Welfare Committee, and the affidavit of permission for animal ethics and welfare’s serial number is No. 20220211.

Camera setting and data acquisition

The videos were recorded using DS-2CD3T47EWDV3-L cameras with a four mm focal length by Hikvision Inc. in Hangzhou, China. 24 h of continuous video data were obtained from 08:00 on 2022.9.17 to 08:00 on 2022.9.18. To monitor the calf feeding behavior in front of the manger from a wide viewpoint, the cameras were positioned in the middle of the fence at a height of 3 m. The layout of the pen and the camera’s placement are shown in Fig. 1.

Figure 1 Layout of the pen and camera setting position.

Feeding behavior definition

As showed in Fig. 2, there are two behavior patterns observed in the range of the trough. First, the behavior occurrs when a yak’s head gets into the rail but its mouth doesn’t touch the feed is named as “picking”. Otherwise, the behavior pattern was defined as “feeding” once the mouth of yak touches the feed. This classification enables a more precise detection of their feeding behavior.

Figure 2 Curve fitting for the feeding and picking dividing.

In order to realize the distinction between feeding and picking behavior in the detection model, the fitting curve of the highest points of feeds in image was used as the boundary of two behavior patterns. As Fig. 2 shows, taking pen-2 as an example, 13 feeds highest points were marked and fitted a curve with python to distinct two behavior patterns in the detection process.

Yak face datasets and feeding behavior datasets

There are two video datasets were made for individual yak detection: yaks’ faces images dataset and yak feeding behavior video dataset.

Yaks’ faces images dataset was prepared for the yak faces classification weight usingYOLOv8. For yaks’ face images dataset, totally 1,243 images of two pens were boxed equally from 11 h video, particular image numbers of each pen were displayed in Table 1. Each yak’s face bounding box was manually labeled with software (LabelImg, 1.8.6; Heartex Inc., San Francisco, CA, USA). The yaks’ faces and ID were displayed in Fig. 3. This study employed a five-fold cross-validation approach to partition the dataset, enabling a comparative analysis of different models’ performance. Subsequently, the entire dataset was utilized for model training to obtain best model parameters.

Table 1 Basic information and data rows of experimental yaks.

Pen number	No. of Yak	Video duration (h)	No. of Yak face images	
1	5	11	579	
2	6	11	664	

Figure 3 Eleven yak face images of two pens.

According to the manual observation, most of feeding behaviors are distribution during 08:30 to 19:30 in this experiment. Thus, yak feeding detection dataset consists of 11 h continuous video of ten yaks raised in two pens collected from 08:30 to 19:30.

Overall individual feeding and picking behavior detection process

PyTorch was used to run the Yolo series and StrongSORT (Rafael Advanced Defense Systems Ltd., Haifa, Israel). Figure 4 illustrates the complete detection procedure from pre-processing to individual yak feeding and picking behavior.

Figure 4 Feeding and picking behavior detection processing flowchart.

From the 24 h continuous yak daily video, 11 h of feeding behavior videos from each pen were selected. Firstly, 579 and 664 yak face images raised in two pens were boxed separately from the videos. The YOLO series models were fed 1,243 images of yak faces in totality to train the weight for the subsequent individual yak detection steps. After that, the StrongSORT detection model used the weight to recognize each yak. In detection, to describe the yak behavior that took place in the trough range precisely, two behavior patterns were identified in this study. Picking is a behavior pattern in which a yak’s head is within range of the trough fence but its mouth does not touch the feed in the trough. Feeding behavior pattern is when a yak lowers its head to graze in the trough. To distinguish two behavior patterns during the detection process, the 13 highest points of feeds were selected and marked in the images, and a curve was fitted to serve as the dividing line between feeding and picking. The detection performance of two pens was tested with 11 h of continuous video, respectively. Every time a yak’s face was detected in the range of the trough as feeding or picking, yak face images, the detected behavior pattern, and the yak’s ID were recorded.

Individual yak feeding behavior detection and feeding time statistic

Spatially enhanced attention module.

A spatially enhanced attention module (SEAM) is a module to improve the classification model’s ability to focus on relevant spatial information within an image. The SEAM module could facilitate multi-scale objects detection, highlighting objects’ regions within images, and weakening the background area oppositely (Yu et al., 2024). The initial component of SEAM is the depthwise separable convolution with residual connection. Depthwise separable convolution is divided channel by channel. To improve inter-channel connectivity, SEAM then employs a two-layer complete connection network to aggregate data from every channel. This framework uses learned associations between unobstructed and unobstructed object regions to compensate in part for information loss caused by occlusion. The fully connected layer’s logits are transformed exponentially to increase the value range from [0, 1] to [1, e]. Finally, the original features are multiplied by the output of the SEAM module allowing the model to effectively manage the yak occlusion. The structure of SEAM module is illustrated in Fig. 5.

Figure 5 Structure of SEAM module.

Individual yak face classification weight training using improved YOLOv8 models.

Training the particular weight of yak face images with yak face classification model is the first step of the individual yak feeding behavior detection. Comparing to the target detection network of the R-CNN series, YOLO improves the model’s running speed while maintaining the detection accuracy essentially unaltered which is suitable for real-time yak detection. The YOLOv8 algorithm is an upgraded version of the YOLOv5 method. This provides technological support for the yak detection model’s real-time performance. The YOLOv8 network model consists of three parts: backbone, neck, and head. The frame structure of improved YOLOv8 is shown in Fig. 6 and the characteristics are described as following:

(1) Backbone: Cross stage partial (CSP) used as the backbone of YOLOv8 (He et al., 2015). Because it splits the feature map into discrete components for convolution operations and their outputs. This strategy not only decreases the model’s computational complexity but also improves the learning capacity of convolutional neural networks. The backbone consists of the CBS, and spatial pyramid pooling fast structures (SPPF). Conv2d, SiLU and BathNorm2d, make up CBS, which is used to adjust the channel count. The performance of feature extraction is significantly impacted by the SPPF module. By pooling features into a fixed-size map, SPPF speeds up computation.

(2) Neck: YOLOv8 uses the C2f module in place of the CSPLayer that was utilized in YOLOv5. the C2f module in YOLOv8 consists of n BottleNeck units and three ConvModule units. This integration into YOLOv8 aims to enhance the model’s ability to capture rich gradient flow information while keeping the algorithm lightweight. By merging contextual data with sophisticated characteristics, the C2f module raises detection accuracy. Besides, the neck network of YOLOv8 also eliminates the convolutional structure from the PAN-FPN upsampling phase present in YOLOv5.

(3) Head: YOLOv8 utilizes an anchor-free model with a decoupled head to perform objectness, classification, and regression tasks independently. The YOLOv8 algorithm incorporates three detection layers in its architecture located within the head, each associated with unique anchors of varying aspect ratios derived from the neck. These layers are utilized for the prediction and regression of targets. Task decomposition enhances the accuracy of the model by enabling it to perform each of the separate jobs more effectively.

The fencing partially obscured the yak’s face as it put its head into the feeding trough. To improve the capacity to detect the yak face in obstructed conditions, the SEAM module described in the ‘Spatially enhanced attention module’ section has been inserted into the head of the YOLOv8 model. Following up-sampling and concatenation procedures, the SEAM module is used to calculate small, medium, and large target feature maps (P3, P4, and P5 layers). The enhancement of the seam module’s yak face identification performance in the case of occlusion was tested by comparing the yak face classification performance by training the yolov8 model with the SEAM module (named “improved YOLOv8 model”) and the original yolov8 model on the same yak face dataset.

Figure 6 The frame structure of improved YOLOv8.

Individual yak face classification weight training using improved YOLOv10 models.

To compare the performance of the YOLOv8 detection models, YOLOv10 is considered as a comparison method. YOLOv10 is latest iteration in the YOLO series which builds upon the advancements of its predecessors. A key innovation in YOLOv10 compare to YOLOv8 is the elimination of the need for non-maximum suppression (NMS) during training. Furthermore, YOLOv10 adopts a holistic model design strategy, optimizing various components for both efficiency and accuracy. Important improvements include lightweight classification heads, spatial-channel decoupled down-sampling, and rank-guided block design (Wang et al., 2024a). The YOLOv10 network model consists of Backbone, Neck, and Head. The frame structure of improved YOLOv10 is shown in Fig. 7 and the characteristics are described as following:

(1) Backbone: the backbone of YOLOv10 employing an advanced version of CSPNet (cross stage partial network) for feature extraction. This enhanced CSPNet architecture is designed to improve gradient flow during training and minimize computational redundancy. YOLOv10 enhances feature extraction capabilities and optimizes the efficiency of the network, thus contributing to more accurate and faster detection performance.

(2) Neck: The neck component of YOLOv10 is pivotal for integrating features across various scales and transmitting them to the head for further processing. The path aggregation network (PAN) layers which are specifically engineered to facilitate effective multiscale feature fusion was employed. The PAN layers aggregate and refine features from different levels of the backbone, thereby enabling the model to leverage both fine-grained and high-level contextual information for improved object detection performance.

(3) Head: The head of YOLOv10 features two key prediction mechanisms: the one-to-many head and the one-to-one head. The one-to-many head generates multiple predictions per object during training and providing diverse supervisory signals that enhance learning and accuracy by allowing the network to capture robust representations and handle object variations effectively. In contrast, the one-to-one head produces a single optimal prediction per object during inference, bypassing the need for non-maximum suppression (NMS) and thereby reducing latency and improving model efficiency, which leads to faster and more streamlined object detection.

The yak face classification performance was also compared by training the yolov10 model with the SEAM module (named “improved YOLOv10 model”) and the original yolov10 model on the same yak face dataset.

Figure 7 The frame structure of improved YOLOv10.

Individual yak detection with StrongSORT.

In this study, StrongSORT was used with the trained particular yak face weight for the individual yak detection. StrongSORT is a multi-object tracking (MOT) method which upgrade from DeepSORT in aspects of detection, embedding and association. StrongSORT’s improvements over DeepSORT are primarily in the two branches (Du et al., 2022). For the appearance branch, BoT (Luo et al., 2020) is used to replace the original simple convolutional neural network (CNN) in DeepSORT. It uses ResNeSt50 (He et al., 2016) as the backbone and pretraining on the DukeMTMC-reID (Ristani et al., 2016) dataset for extracting more discriminative features. Furthermore, feature bank is replaced with the feature updating strategy proposed in Wang et al. (2020), which updates appearance state eit for the ith tracklet at frame t in the following way: (1) eit=αeit−1+1−αfit

where fit is the current matched detection’s appearance embedding and 0.9 is a momentum term. The EMA updating strategy improves matching quality and running speed.

For the motion branch, StrongSORT select ECC (Evangelidis & Psarakis, 2008) for camera motion compensation. NSA Kalman algorithm is adopted instead of vanilla Kalman filter. Here is a formula to adaptively calculate the noise covariance Rk: (2) Rk ˜=1−ckRk

where Rk is the preset constant measurement noise covariance and ck isthe detection confidence score at state k.

In addition, to solve the assignment problem in both appearance and motion information, Cost matrix C is a weighted sum of appearance cost Aa and motion cost Am, as shown below: (3) C=λA_a+1−λA_m

where weight factor λ is set to 0.98.

Individual feeding and picking distribution and duration evaluation method

The distribution and duration of feeding and picking behavior were displayed and evaluated in order to present behavior habits during feeding of individual yaks in a more intuitive and comprehensive way. To better demonstrate the feeding and picking behavior, we did not print the yak’s position at every single frame, instead printing each dot every 12 s, as shown in Fig. 8. Thus, the feeding and picking behaviors were observed to be clearly dependent on the yak’s position in the Fig. 8. Furthermore, the real-time behavior detection of yaks was executing every three frames. And each second contains 25 frames, duration (minutes) of each behavior pattern is calculated using the formula: (4) tb=d×325×60

where tb represents the evaluated duration of each behavior, and d represents the total number of times the detection model counted for each behavior.

Figure 8 Feeding and picking behavior distribution of Yak-1 during one day.

Experimental platform

The experiment was conducted on an Ubuntu 18.04 operating system, equipped with an AMD Ryzen 9 5950X CPU and two NVIDIA GeForce RTX 3090 GPUs. The PyTorch framework (version 1.8) and Python language (version 3.9) were utilized for the experiment. The main parameter settings of training are shown in Table 2. In the tracking period, the reid-weights were set to osnet_x0_25_msmt17.pt, with the confidence threshold and IoU threshold both set at 0.5.

Table 2 Parameters setting for training.

Parameters	Batch Size	Epochs	Optimizer	Learning rate 0	Learning rate f	Image size	
Value	32	300	SGD	0.01	0.01	640 × 640	

Model evaluation

The overall performance of the detection model is represented by accuracy, precision, recall, mean average precision 0.5 (mAP0.5) and mAP0.5:0.95. The detection results were manually checked, and the case was considered “positive”. If the yak’s face image, behavior pattern, and ID were consistent with the actual situation. Otherwise, mistaking one yak’s face for another was considered a “negative class”. When the classification was analyzed, the models accurately predicted the positive and negative classes, referred to as “true positive” and “true negative”, respectively. False positives and false negatives are model predictions that are incorrect for the positive and negative classes, respectively. The terms accuracy, recall, precision, and F1-score are defined as follows:

(5) accuracy=TP+TNTP+TN+FP+FN

(6) recall=TPTP+FN

(7) recall=TPTP+FN

(8) F1-score=2TP2TP+FP+FN

where define TP (True positive) and TN (True negative), respectively. Incorrect model prediction of the positive and negative classes are defined as FP (False positive) and FN (False negative).

mAP0.5 is defined as the average precision at an IoU threshold of 0.5. Meanwhile, mAP0.5:0.95 represents the average precision across IoU thresholds ranging from 0.5 to 0.95, with increments of 0.05. The corresponding formulas are as follows:

(9) mAP0.5=1C∑j=1CAPj

(10) mAP0.5:0.95=110∑t=0.50.95mAPt

where, APj reflects the model’s ability to predict objects of class j accurately at 0.5 IoU thresholds. mAPt is the average precision at a specific IoU threshold.

Results

Yak face detect model performance

In this study, four YOLO series networks trained on labelled images of individual yak faces were used to detect individual feeding behaviors of eleven yaks and fold cross-validation was used to evaluate the classification performance. The average value of the result metrics across the five folds were calculated as the evaluation indicator for the models. The yak face classification performance of YOLOv8, improved YOLOv8, YOLOv10, and the improved YOLOv10 are shown in Table 3. Overall, the improved YOLOv8 model achieved the best performance with precision, recall, mAP50, and mAP50-95 of 96.78%, 96.30%, 97.59%, and 79.60%, respectively. The utilization of the SEAM module in both YOLOv8 and YOLOv10 models resulted in improvements of recall, mAP50, and mAP50-95.

Table 3 Comparison of the detection results of four models.

Model	Model performance indicators		
	Precision	Recall	mAP50	mAP50-95	GFLOPs	
YOLOv8	96.57%	96.03%	97.47%	79.42%	8.1	
Improved YOLOv8	96.78%	96.30%	97.59%	79.60%	8.5	
YOLOv10	96.51%	96.08%	97.45%	79.31%	8.2	
Improved YOLOv10	96.09%	96.57%	97.69%	79.39%	8.7	

Yak feeding and picking behavior detection model performance

The overall detection performance of yak’s feeding and picking behavior is showed in Table 4 and Fig. 9 displays an example of the feeding and picking behavior detection results. Using the improved YOLOv8 model for tracking feeding behavior, the detection model had higher accuracy, precision, recall, and F1-score results of 98.76%, 98.77%, 98.68%, and 98.72%, respectively.

Table 4 Individual feeding and picking behavior detection performance of two pens.

Model	Model performance indicators	
	Accuracy [%]	Precision [%]	Recall [%]	F1-score [%]	
YOLOv8	98.23	98.16	98.12	98.14	
Improved YOLOv8	98.76	98.77	98.68	98.72	
YOLOv10	98.66	98.61	98.57	98.59	
Improved YOLOv10	97.78	97.69	97.82	97.75	

Figure 9 An example of improved YOLOv8 feeding and picking behavior detection results.

The individual feeding and picking behavior detection confusion matrices of four YOLO series model are shown in Fig. 10. In all behavior detection models, Yak-4, Yak-6, Yak-9 were classified with a relatively high accuracy which are over 99%. Specifically, the detection accuracy of Yak-4 is exceeding 99.2%. The major mistake in the feeding and picking behavior detection process was the subsequent behavior misdetection caused by the misclassification of the individual yak face. Yak-1 was mainly misclassified as Yak-3 among all models, with a ratio from 1.3% to 4.9%. For the detection of Yak-2, 0.9% to 1.7% yak face images were misidentified as Yak-5. Feeding behavior detections of yak-5 were-relatively equally recognized as Yak-1, Yak-2, Yak-3, and Yak-4 from 0.1% to 0.8% of images. As observed in detection results of Yak-11, the improved YOLOv8 and improved YOLOv10 model had the best classification accuracy of 99.4% and 99.6% respectively, comparing to the original YOLO series models.

Figure 10 (A–D) Confusion matrix of the feeding and picking detection results.

Yak feeding and picking behavior distribution and lasting time statistic

Based on the performance of the two models described in ‘Yak feeding and picking behavior detection model performance’, individual yak face weight trained with YOLO series were used to detect feeding and picking behaviors in this study. Thus, based on individual behavior pattern detection, the distribution and duration of each yak can be computed and visualized, and the one-day statistical results are shown in Fig. 11. As shown in Fig. 11, feeding and picking behavior is primarily distributed during two times: 8:30 a.m. to 10:30 a.m. and 17:00 p.m. to 19:30 p.m., which corresponds to actual feed delivery timing. All yaks’ feeding behavior duration is longer and more concentrated when compared to picking behavior. While there are similarities in the feeding and picking behaviors of 11 yaks, there are also individual differences in the feeding period. The majority of yaks ate with focus after delivery; however, Yak-1 and Yak-6 fed without focus for a longer period of time. In addition, Yak-2 didn’t eat while the other yaks were trying to eat. According to the reference in the video, Yak-2 cannot be fed on a regular time because she is attacked by Yak-5 on this particular day.

Figure 11 (A–B) Feeding and picking behavior distribution of 11 yaks during one day.

Furthermore, the feeding and picking behaviors duration in one day were calculated based on individual behavior detection. Figure 12 represents the feeding and picking behavior duration of each yak corresponding to Fig. 3. Feeding duration are clearly longer than picking duration in general, which is consistent with the behavior-distribution situation. The feeding and picking behavior distribution and duration provide an intuitive visualization of yak’s individual feeding habit.

Figure 12 (A–D) Feeding and picking behavior lasting time statistic during day time.

Overall, the four models show a relatively high degree of consistency in the duration statistics trends as shown in Fig. 12. Table 5 compares the duration statistics conducted by each model with the actual durations. Among these models, the improved YOLOv8 and StrongSORT model achieved the smallest overall average error value 4.75.

Table 5 Mean absolute error (MAE) of four models for feeding and picking behavior duration.

Model	YOLOv8_ StrongSORT	ImprovedYOLOv8_ StrongSORT	YOLOv10_ StrongSORT	ImprovedYOLOv10_ StrongSORT	
Overall MAE	5.6	4.75	5.62	7.36	

Discussion

Comparison with the state-of-the-art

An individual yak feeding and picking behavior detection model with StrongSORT was developed in this research using the yak face classification weight trained by YOLO series. Livestock animal, including yaks, develops its own distinct feeding habits based on genetics and the environmental elements. However, existing methods rarely combined real-time identity authentication with individual feeding behavior detection (Tassinari et al., 2021; Myat Noe et al., 2023). Current research primarily achieves the monitoring of the feeding behavior of cows, by using sound sensors, accelerometer sensors, and IMU sensors. Usually, these studies achieve better monitoring results. The challenge with accelerometer and IMU sensors is that these wearable equipments have an impact or stress on livestocks’ daily activities (Seneviratne et al., 2017; Yin et al., 2023). In this study, images and videos collected from 11 yaks to train the yak face classification YOLO series models and the StrongSORT detection model. As shown in Table 6, compared with models in previous researches, the present study achieved better results in evaluation indicators such as accuracy, precision, and F1-score.

Table 6 Feeding behavior classification accuracy of the developed model from the present study compared with past models.

Work	Sensors	Model	Results	
Martinez-Rau et al. (2024)	Sound sensor	The chew–bite energy-based algorithm	F1-score: 87.0%.	
Benaissa et al. (2019)	Accelerometer sensors	SVM model	Precision: 92%.	
Rahman et al. (2018)	Accelerometer sensors	Random Forest model	F1-score: 80.90%.	
Liu et al. (2023)	IMU sensors	fully convolutional network model	Accuracy: 92.8%	
Zhang et al. (2023)	IMU sensors	LSTM model	Precision: 91.05%.	
Present study	2D image	Improved YOLOv8+StrongSORT	Accuracy: 98.76%; precision: 98.77%; recall: 98.68%; f1-score: 98.68%.	

General discussion

In this study, the improved YOLOv8 model achieves the best detection performance in all evaluation indicators including accuracy, precision, recall, mAP0.5 and mAP0.5:0.95. In general, two YOLOv8 models show better detection performance compared to two YOLOv10 models. The reason of the superior performance of the YOLOv8 models are attributed to the differences in their feature extraction modules. In the situation of yak faces obscured by the fences, the utilization of SEAM module enhances multi-scale object detection by focusing on object regions within images while suppressing background noise. This improves the detection capability of YOLOv8 and YOLOv10 in crowing and occlusion conditions.

Few yak faces were misidentified as other yaks in individual feeding and picking behavior detection tasks. Three factors contributed to these incorrect detections. To begin, as stated in the results, yak faces with similar features, such as a white spot on their faces, have a chance of being confused with one another. Most Yaks have various patterns of white spots on their faces. Yak-4, which is entirely black, achieves the highest recognition accuracy. Yak-11, with a black face and a grey mouth, also shows relatively high accuracy. Furthermore, because of crowding or social interaction, the yaks’ faces are sometimes obscured by the heads of other yaks. A portion of Yak-11’s face was incorrectly identified as Yak-8. Manual verification revealed that these misidentifications occurred during a specific period when Yak-11 was captured from a side profile during social interactions, which caused the detection models to fail in recognizing all facial features accurately. Besides, when the staff delivered the food, the staff’s body covered part of the yak’s face, some facial features were not recognized by the detection model.

To improve detection performance in above situations, more angles and conditions, such as side yak faces and sheltered yak faces, should be included in the yak faces images dataset. Furthermore, images collected in various weather conditions, light conditions, and cultivation environments should be considered in the dataset preparation to develop the model’s robustness.

Conclusion

The feeding and picking behaviors of yaks were defined in this study. YOLO series and StrongSORT models were used to detect the feeding and picking behaviors of 11 yaks raised. The improved YOLOv8 and StrongSORT model achieved the best performance, and the detection accuracy, precision, recall, F1-score was 98.76%, 98.77%, 98.68% and 98.72% respectively. In addition, the feeding behavior, picking distribution, and lasting time statistics of each yak were computed and visualized based on the automatic individual behavior detection. Real-time individual feeding and picking behavior detection is achieved through combining the yak faces classification and multiple yak detection tasks. As a result, a potential technology for monitoring and visualizing yaks’ feeding behavior was developed. It can also be utilized with other animals, such as cattle and sheep housed in pens. In the future, a dataset of yaks in varied cultivate conditions, group sizes, and lighting circumstances will be included. For the purpose of predicting livestock weight, the association between feeding interval and yak weight gain will also be explored in the future.

Supplemental Information

Supplemental Information 1 Video 1

Supplemental Information 2 Video 2

Supplemental Information 3 Video 3

Additional Information and Declarations

Competing Interests

Author Contributions

Ethics

Data Availability

The authors declare there are no competing interests.

Yuxiang Yang conceived and designed the experiments, performed the experiments, analyzed the data, performed the computation work, prepared figures and/or tables, authored or reviewed drafts of the article, and approved the final draft.

Meiqi Liu conceived and designed the experiments, performed the experiments, analyzed the data, performed the computation work, prepared figures and/or tables, and approved the final draft.

Zhaoyuan Peng conceived and designed the experiments, performed the experiments, analyzed the data, performed the computation work, prepared figures and/or tables, and approved the final draft.

Yifan Deng conceived and designed the experiments, performed the experiments, performed the computation work, authored or reviewed drafts of the article, and approved the final draft.

Luhui Gu performed the experiments, analyzed the data, authored or reviewed drafts of the article, and approved the final draft.

Yingqi Peng conceived and designed the experiments, performed the experiments, analyzed the data, performed the computation work, prepared figures and/or tables, authored or reviewed drafts of the article, and approved the final draft.

The following information was supplied relating to ethical approvals (i.e., approving body and any reference numbers):

The experiment was assessed by the Sichuan Agricultural University Institutional Animal Care and Use Committee, and the affidavit of permission for animal ethics and welfare’s serial number is No.20220211.

The following information was supplied regarding data availability:

The running code is available at Zenodo: zhengzhule. (2024). zhengzhule/feeding-behavior-detection: code to feeding behavior detection (v1.0.0). Zenodo. https://doi.org/10.5281/zenodo.13683772.

The dataset used for training cowface is available at Zenodo: zhengzhule. (2024). zhengzhule/data: data to feeding behavior detection (v1.0.0). Zenodo. https://doi.org/10.5281/zenodo.13683891.

The raw video data are available in the Supplemental Files.

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
