# Peer review of "A real-time feeding behavior monitoring system for individual yak based on facial recognition model"

_PeerJ Computer Science, doi:10.7717/peerj-cs.2427_

## Round 0.1 · original submission · Major Revisions

Please revise the paper according to the reviewer's comments.

Reviewer 1 ·

Basic reporting

All comments have been added in detail to the last section.

Experimental design

All comments have been added in detail to the last section.

Validity of the findings

All comments have been added in detail to the last section.

Additional comments

Review Report for PeerJ Computer Science
(A real-time feeding behavior monitoring system for individual yak based on facial recognition model)

1. Within the scope of the study, various classification and detection studies were conducted using videos and images containing yak face images with deep learning methods.

2. Since the yak images used in the study were collected specifically for the study, the study has originality in terms of dataset.

3. The introduction section needs to be detailed more in terms of yak images or studies in similar fields. In addition, it is very important to make the difference of the study from the literature and the main contributions to the literature more explanatory in items in order to clearly emphasize the importance of the study, its originality and the importance of the subject.

4. Regarding the problem solution, it is stated that although there are many different deep learning-based models in the literature, yolo and StrongSORT were used in this study. Explain the reason why v8 was preferred among the Yolo versions. The latest updated yolo version is v10. In addition, there are many different object detection models that can be used other than yolo. In terms of comparing the results, adding a few state-of-the-art deep learning models has the potential to add depth to the study.

5. It is stated that the dataset for yak face classification is divided into 70% and 30% as training and testing, respectively. Since the results of the classification problems are very dependent on the dataset, how the dataset distribution is made and the reasons are very important. For this reason, explain in detail the reasons for the dataset percentage selection and/or the reasons for not preferring cross-validation, which is frequently used in the literature to reduce dataset dependency and increase the reliability of the results.

6. More details should be given regarding the hyperparameters of the deep learning models used. In addition, although it is stated that PyTorch is used as the program, more details should be given regarding the toolbox/framework used for the program.

7. When the sample image in Figure-7 is examined, it is observed that yak face detection operations are performed. However, there are serious deficiencies in many evaluation metrics, especially regarding object detection in the results.

As a result, although the study is important in terms of dataset usage and the subject discussed, the sections mentioned above should definitely be clarified.

Reviewer 2 ·

Basic reporting

In this paper, the authors presented a real-time detection model for individual yak feeding and foraging behavior using YOLOv8 and StrongSORT. The authors used images and videos collected from 11 yaks raised in two pens to train the yak face classification YOLOv8 models and the StrongSORT detection model. The yak behavior patterns detected in the trough area were defined as feeding and picking, and the overall detection performance of these two behavior patterns was described using metrics such as accuracy, precision, recall, and F1 score.

My comments:

- The introduction is poorly written and needs rewriting.

- The contributions of the work are not clearly mentioned.

- Some acronyms are not defined.

- Parameters in equations 1-4 are not defined.

Experimental design

- The experimental data set is unknown (self-made) and the results cannot be reproduced.

- There is no comparison with state-of-the-art methods under the same conditions (i.e., data set and experimental protocol).

Validity of the findings

- The authors should report their main findings in the abstract and introduction sections.

- No real insight into the relevant studies has been provided.

- The literature gap is not well defined!

Additional comments

- Most of the references are outdated, the authors are requested to update the references.

Reviewer 3 ·

Basic reporting

This manuscript presents an interesting study on using advanced technology to monitor yak feeding behaviors in feedlots. The use of YOLOv8 and StrongSORT for real-time detection seems promising, especially with high accuracy rates in different pens.
The manuscript is well written and organized However, I have several comments to further improve the quality of the manuscript:
- Authors should outline the major contributions of the study as bullet points at the end of the introduction to immediately clarify what novel or significant aspects the study brings to the field. Similarly, organizing the sections within the introduction helps the reader understand the structure and flow of the article.
- The transition to the specific objective of the study (developing a method to detect individual feeding behavior in yaks) is clear. However, it would be helpful to briefly explain in introduction why classifying feeding behaviors into "feeding" and "picking" is important for fully understanding yak feeding habits.

Experimental design

- Using manual checks to correct detection errors can introduce potential biases, especially if the samples are larger. What do you think could be done to address this issue?

Validity of the findings

- Authors should include a 'Related Work' section discussing the compared methods presented in Table 3.

Additional comments

- Authors should choose to use either "Figure" or "Fig." consistently for all figures in their publication to ensure a uniform and clear presentation of visual information example: ”Furthermore, the feeding and picking behaviors duration in one day were calculated based on individual behavior detection. Figure 10 represents the feeding and picking behavior duration of each yak corresponding to Fig. 9.”
- Authors should indeed verify the clarity and readability of textual content in images and tables.

---

## Round 0.2 · accepted · Accept

According to the comments of reviewers, after comprehensive consideration, it is decided to accept it.

Reviewer 1 ·

Basic reporting

All comments have been added in detail to the last section.

Experimental design

All comments have been added in detail to the last section.

Validity of the findings

All comments have been added in detail to the last section.

Additional comments

Thanks for the revision. The responses and changes made to the article based on the reviewer comments are sufficient. I recommend that the paper be accepted.

Reviewer 2 ·

Basic reporting

The manuscript has been greatly improved and can be accepted in the current form.

Experimental design

/

Validity of the findings

/

Additional comments

/

Reviewer 3 ·

Basic reporting

The article is now approved and can be published.

Experimental design

/

Validity of the findings

/

Additional comments

/